# The association between maternal prenatal folic acid and multivitamin supplementation and autism spectrum disorders in offspring: An umbrella review

Biruk Beletew Abate[1,2]*, Biruk Shalmeno Tusa[1,3], Ashenafi Kibret Sendekie[4,5], Desie Temesgen[6], Kindie Mekuria[6], Addis Wondmagegn Alamaw[6], Molla Azmeraw[6], Alemu Birara Zemariam[6], Tegene Atamenta Kitaw[6], Amare Kassaw[7], Gebremeskel Kibret Abebe[6], Befkad Derese Tilahun[6], Gizachew Yilak[6], Molalign Aligaz Adisu[6], Berihun Dachew[1,8]

1 School of Population Health, Curtin University, Perth, Washington Australia, 2 College of Medicine and Health Sciences, Woldia University, Woldia, Ethiopia, 3 Department of Epidemiology and Biostatistics, College of Health and Medical Sciences, Haramaya University, Haramaya, Ethiopia, 4 Department of Clinical Pharmacy, School of Pharmacy, College of Medicine and Health Sciences, University of Gondar, Gondar, Ethiopia, 5 School of Pharmacy, Curtin Medical School, Faculty of Health Sciences, Curtin University, Bentley, Washington, Australia, 6 Department of Nursing, College of Health Science, Woldia University, Woldia, Ethiopia, 7 Department of Nursing, College of Health Science, Debre Tabor University, Ethiopia, 8 enAble Institute, Curtin University, Perth, Western Australia, Australia

* birukkelemb@gmail.com

## Abstract

### Background

Previous reviews have examined the association between maternal prenatal use of folic acid and multivitamin supplements and autism spectrum disorder (ASD) in children, but findings remain inconclusive. This umbrella review aims to synthesise the existing evidence on the association between prenatal folic acid and multivitamin supplementation and the risk of ASD in offspring.

### Methods

This umbrella review followed the PRISMA guidelines to synthesise and report evidence from existing systematic reviews and meta-analyses (SRMs). Articles were searched in PubMed, Scopus, Web of Science, and Google Scholar. The quality of included studies was assessed using the Assessment of Multiple Systematic Reviews (AMSTAR). A weighted inverse variance random-effects model was applied to estimate pooled effects. The association was quantified using relative risks (RRs) with 95% confidence intervals (CIs). Subgroup analysis and sensitivity analysis were also conducted. Heterogeneity and publication bias were assessed.

**Data availability statement:** Data are all contained within the paper and/or Supporting Information files.

**Funding:** The author(s) received no specific funding for this work.

**Competing interests:** The authors have declared that no competing interests exist.

## Results

Eight SRMs comprising 101 primary studies and over three million mother-offspring pairs were included. Prenatal folic acid and/or multivitamin supplementation was associated with a 30% reduced risk of ASD in offspring (RR = 0.70, 95% CI: 0.62, 0.78; GRADE: highly suggestive). Subgroup analysis by supplement type showed that maternal prenatal multivitamin supplementation reduced the risk of ASD by 34% (RR = 0.66, 95% CI: 0.55–0.80; GRADE: highly suggestive), while folic acid supplementation was associated with a 30% reduction in ASD risk (RR = 0.70, 95% CI: 0.60–0.83; GRADE: highly suggestive).

## Conclusion

Maternal prenatal folic acid and multivitamin supplementation are associated with a reduced risk of ASD in offspring. These findings have important public health implications, suggesting that prenatal supplementation could help mitigate the risk of ASD in children.

## Background

Autism spectrum disorder (ASD) is a pervasive neurodevelopmental disorder that has an impact on reciprocal social interactions, nonverbal communication, and understanding of social relationships [1]. It is frequently associated with co-occurring illnesses such as epilepsy, depression, anxiety, and attention deficit hyperactivity disorder, as well as problematic behaviours such as trouble sleeping and self-injury [1,2].

Autism spectrum disorder is a global public health problem affecting up to 1% of children around the world [3]. The etiology and risk factors have been proposed to a single or a group of genetic mutations and environmental influences [4]. Ongoing studies are most frequently environmental risk factors [5], and prenatal maternal nutrition are among the modifiable risk factors for ASD [6]. One of the maternal nutritional factors that may reduce the risk of offspring ASD is prenatal folic acid and multivitamin supplementation. Folic acid supports DNA methylation and epigenetic regulation, both critical for neurodevelopment, and aids in neural tube formation, protecting against defects associated with neurodevelopmental disorders [7,8]. Multivitamins, which typically contain essential nutrients like vitamin B12, vitamin D, and iodine, help maintain immune system balance, reduce inflammation, and support brain function by ensuring proper neurotransmitter synthesis and amino acid metabolism [9]. These nutrients contribute to optimal fetal brain development, potentially lowering the risk of ASD.

Early prevention is preferable to treatment, as no medication can address the core symptoms of ASD; available treatments focus on managing specific co-occurring conditions or symptoms. Maternal supplementation with folic acid and multivitamins before and during pregnancy represents one of the most accessible and

cost-effective preventive strategies. To date, eight systematic reviews and meta-analyses (SRMs) comprising 101 primary studies have examined whether maternal prenatal folic acid supplementation reduces the risk of ASD in offspring. However, findings from previous reviews remain inconclusive. While some reviews have reported the association between prenatal folic acid and multivitamin supplementation and a lower risk of ASD, others have found no association. However, findings remain inconclusive due to several limitations in the existing literature, including variability in study design and population characteristics, differences in the definition and measurement of supplementation exposures (folic acid alone versus multivitamins, timing, dose, and duration), heterogeneity in ASD diagnostic methods (ranging from clinical diagnoses to parent-reported outcomes), and potential publication or reporting bias. These limitations have contributed to conflicting findings: while some SRMs report a protective effect of maternal folic acid and/or multivitamin supplementation [10–14], others find no significant association [15,16]. These inconsistencies underscore the need for a more comprehensive synthesis. An umbrella review, unlike a single SRM, offers a higher-level overview by integrating findings across multiple SRMs, critically appraising their methodological quality, and grading the overall strength of the evidence. Accordingly, this umbrella review aimed to synthesise these inconsistent findings to clarify the potential role of prenatal folic acid and multivitamin supplementation in reducing the risk of ASD in offspring.

## Methods

### Research design and searching strategy

This umbrella review was conducted following the Preferred Reporting Items for Systematic Reviews and Meta-Analyses (PRISMA) guidelines [17]. Additionally, to ensure the study's rigor, we adhered to the PRIOR statement, a reporting guideline for overviews of reviews on healthcare interventions [18].

Articles were searched across four databases including PubMed, Scopus, Web of Science, and Google Scholar. Additionally, a complementary search was conducted in Google Scholar to capture grey literature sources. We performed a comprehensive systematic search for SRMs that assessed the association between prenatal folic acid and/or multivitamin supplementation and the risk of ASD on a global scale, using the PICO (Population, Intervention, Comparison, and Outcome) framework. The search strategy incorporated MeSH terms, keywords, and their combinations. Moreover, we manually reviewed references cited in relevant systematic reviews and conducted a snowball search to identify linked articles from these studies. The search terms included: (Preconception OR pre-conception OR prenatal) AND (Multivitamin OR vitamin OR mineral OR micronutrient OR antioxidant OR diet OR folic acid OR folic-acid) AND (autism OR autistic OR ASD OR Asperger OR pervasive developmental disorder OR PDD) AND (meta-analysis OR systematic review OR review).

### Study selection/eligibility criteria

The retrieved SRMs were imported into EndNote 20 software. Screening followed a two-stage process: title and abstract screening, followed by full-text review. Two independent investigators (BB and MA) applied pre-specified inclusion criteria to identify relevant articles. Eligible studies met the following criteria: [i] had a defined literature search strategy, [ii] appraised the included studies using a relevant tool, and [iii] used a standard approach to pooling data and providing summary estimates. Studies were excluded if they lacked relevant outcome measures, were not in English, or were narrative reviews, editorials, correspondence, conference abstracts, or methodological studies. Disagreements in both title/abstract and full-text screening were addressed through structured discussions during consensus meetings, where team members collaboratively reviewed the criteria and reached a resolution based on predefined guidelines.

### Criteria for considering studies for this umbrella review

**Types of studies.** All SRMs-analyses assess the association between maternal prenatal multivitamins and/or folic acid supplementation and ASD in the offspring.

**Types of participants.** Participants who have taken folic acid and/or multivitamin supplementation during the prenatal period, regardless of whether the pregnancy was single or multiple, and irrespective of their gestational age, were considered.

**Types of interventions.** Folic acid and multivitamin supplementation were administered to women before pregnancy and compared with a placebo, no treatment, or an alternative agent to improve outcomes. We included studies where different regimens for helping folic acid and multivitamins were compared.

Outcomes of interest: The incidence of ASD among intervention groups (whose mothers have taken folic acid and multivitamins) compared to the control groups (whose mothers didn't take folic acid and multivitamins) was estimated.

**Setting:** Global

**Publication condition**: Published articles that reported the effect of the outcome of interest

## Quality assessment

The methodological quality of all included reviews was assessed by two independent reviewers using the Assessment of Multiple Systematic Reviews (AMSTAR) tool [19,20]. The quality was scored out of 11, with scores < 3, 4–7, and 8–11 indicating low, moderate, and high qualities, respectively. The decision as to whether to include a review was made based on meeting a pre-determined proportion of all criteria, or on certain criteria being met. Decisions about a scoring system or any cut-off for exclusion were made in advance and agreed upon by all reviewers before critical appraisal commences.

## Data extraction

Data from the included SRM studies were extracted using a standardized data abstraction form, developed in an Excel spreadsheet. For each SRM study, the following data were extracted: (a) identification data (first author's last name and publication year), (b) Review aim and type, (c) measure of association (odds ratio or relative risk with 95% confidence intervals), (d) number of primary studies included within each SRM study and their respective design type, (e total number of sample size included, (f) publication bias assessment methods and scores, (g) quality assessment methods and scores, (h) and data synthesis methods (random or fixed-effects model) [Table 1].

## Statistical analysis

STATA v17.0 software was used for statistical analyses. Both narrative and qualitative methods, including tables and forest plots, were employed to summarize the estimates from the included reviews. We conducted an inverse variance-weighted random-effects meta-analysis to estimate pooled effect sizes with 95% CIs [21]. We assessed heterogeneity across studies using Cochrane's Q test (Chi-square), the $I^2$ statistic, tau-squared ($\tau^2$), and corresponding p-values [22]. Heterogeneity was interpreted according to conventional thresholds, with $I^2$ values of 0% indicating no heterogeneity

**Table 1. Characteristics of included studies.**

| Sr No | Authors | Year | Study design | Number of articles included | Sample size | Supplement | Timing | Estimate (95%CI) |
|---|---|---|---|---|---|---|---|---|
| 1. | Guo B-Q, et al.[10] | 2019 | SRM | 5 | 231,163 | Multivitamin | Prenatal | RR = 0.62 (0.45, 0.86) |
| 2. | Guo B-Q, et al. [15] | 2019 | SRM | 8 | 840,776 | Folic acid | Prenatal | OR = 0.91 (0.73,1.13) |
| 3. | Li M, et al.[11] | 2019 | SRM | 20 | 1038013 | Multivitamin | Prenatal | RR = 0.64 (0.46, 0.9) |
| 4. | Friel C, et al.[16] | 2021 | SRM | 10 | 904,947 | Multivitamin | Prenatal | RR = 0.74 (0.53, 1.04) |
| 5. | Liu X, et al. [12] | 2021 | SRM | 10 | 9795 | Folic acid | Prenatal | OR = 0.57 (0.41, 0.78) |
| 6. | Wang M, et al.[27] | 2017 | SRM | 16 | 4514 | Folic acid | Prenatal | OR = 0.77 (0.64,0.93) |
| 7. | Iglesias VL et al [13] | 2018 | SRM | 16 | 756,365 | Folic acid | Prenatal | OR = 0.58 (0.46, 0.75) |
| 8. | Chen H et al.[14] | 2023 | SRM | 15 | 1,116,337 | Folic acid | Prenatal | OR = 0.68 (0.53,0.83) |

and values of 25%, 50%, and 75% representing low, moderate, and high heterogeneity, respectively [20,23]. For analyses where significant heterogeneity was detected, we applied the DerSimonian–Laird random-effects model. Subgroup analyses were conducted according to supplement type (folic acid versus multivitamin). To evaluate the robustness of findings, sensitivity analyses were performed by sequentially excluding individual studies. Publication bias was assessed using both graphical and statistical approaches: visual inspection of funnel plot symmetry and Egger's regression test. Funnel plot asymmetry and a p-value <0.05 in Egger's test were considered indicative of potential publication bias [24].

### Credibility/GRADE assessment of each pooled analysis

We used the GRADE system to assess the strength of evidence for each pooled analysis, classifying it as convincing, highly suggestive, suggestive, or weak. GRADE uses five criteria: risk of bias, inconsistency, indirectness, imprecision, and publication bias. Downgrades occur based on a high risk of bias, inconsistency (high I² values), indirectness (multiple control comparisons), imprecision (wide confidence intervals or small sample size), and publication bias (significant Egger test p-value). Convincing evidence (Class I) required a $p\text{-value} < 10^{-6}$, over 1,000 participants, low-to-moderate heterogeneity ($I^2 < 50\%$), a 95% prediction interval excluding the null, and no small-study bias. Highly suggestive associations (Class II) involved over 1,000 participants, a $p\text{-value} < 10^{-6}$, and a 95% prediction interval excluding the null. Suggestive evidence (Class III) required over 1,000 participants and a $p\text{-value} \leq 0.001$, while weak associations (Class IV) needed a $p\text{-value} \leq 0.05$, with $p > 0.05$ considered non-significant [25,26].

### Ethics approval and consent to participate

This study is a systematic review of published articles and does not require ethical approval.

## Results

A total of 1388 reviews were identified; 1380 were from different databases, and eight were from manual sources. After duplicates were removed, 667 articles remained to be screened for title and abstract. After title and abstract screening, 142 studies remained for full-text review. Finally, after excluding 134 (using eligibility criteria), 8 SRMs, comprising 101 primary studies with 3,029,208 participants were included in the final analysis (Fig 1).

### Characteristics of included studies

All the included reviews were both systematic reviews and meta-analyses [10–16,27]. Five of the included SRMs examined the effect of maternal folic acid supplementation [12–15,27], while the other three investigated the impact of multivitamin supplementation [10,11,16] on ASD in offspring. The included SRMs range from five [14] to 20 [15] primary studies. The minimum and maximum sample sizes in these included SRM were 4,514 [27] and 1,090,585 [14] respectively (Table 1).

### Quality assessment

Based on AMSTAR-2 criteria, the majority of the SRMs (6 out of 8) [10,12,13,15,16,27] were assessed as high quality, while 2 out of 8 were rated as low quality [11,14] (S1 Table). The most common reasons for a low-quality rating based on the AMSTAR-2 checklist are: failure to register the protocol in PROSPERO [10–13,15,27], absence of an evaluation of the risk of bias in the meta-analysis [10,11,13–15,27], insufficient explanation of heterogeneity [11,16], and failure to assess publication bias [10,11] (S1 Table).

### Effect of maternal prenatal folic acid and multivitamin supplementation on offspring ASD

Among the eight studies included, six reviews reported an association between prenatal folic acid and multivitamin supplementation and a reduced risk of ASD [10–14], whereas the remaining two reviews, one for folic acid and one for multivitamin,

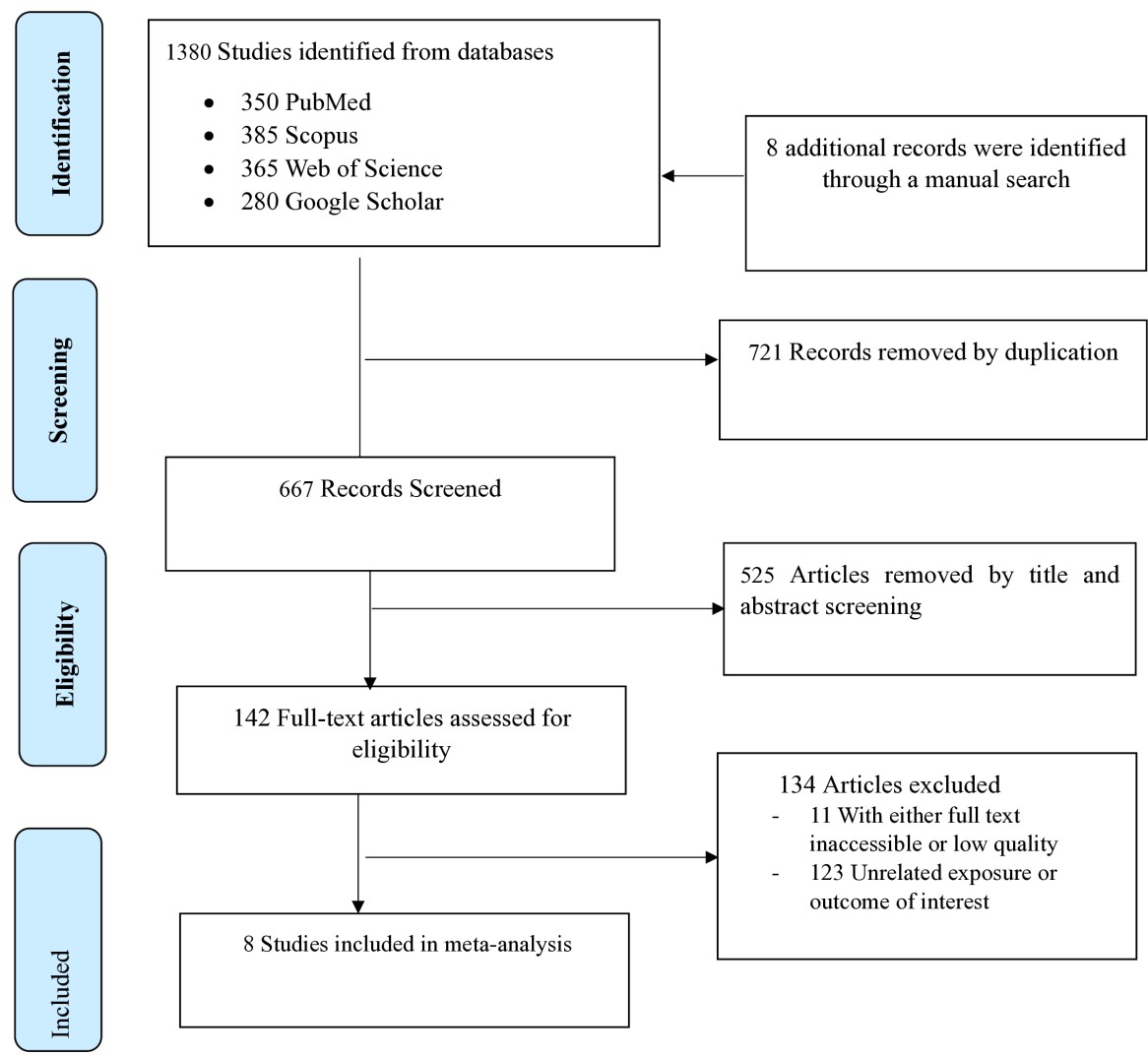

**Fig 1. PRISMA flow diagram shows the search results and reasons for the exclusion.**

found no such association [15,16]. The effect estimates from these included studies ranges from 0.57 (95% CI: 0.41, 0.78) [12] to 0.91 (95% CI: 0.73, 1.13) [10].The random-effects model analysis revealed a 30% reduced risk of ASD (RR=0.70, 95% CI: 0.62–0.78) in offspring of mothers who took folic acid and/or multivitamin supplementation during prenatal care, compared to those who did not. Subgroup analysis by supplement type showed that maternal prenatal multivitamin supplementation reduced the risk of ASD by 34% (RR=0.66, 95% CI: 0.55–0.80; GRADE: highly suggestive), while folic acid supplementation was associated with a 30% reduction in ASD risk (RR=0.70, 95% CI: 0.60–0.83; GRADE: highly suggestive) (Fig 2).

## Sensitivity analysis

The sensitivity analysis indicated that any of the single studies did not influence the pooled result. The pooled sensitivity estimate for the association between maternal prenatal folic acid and multivitamin supplementation and offspring ASD ranged from 0.67 (95% CI: 0.61–0.74) [15] to 0.72 (95% CI: 0.64–0.81) [13] (S1 Fig).

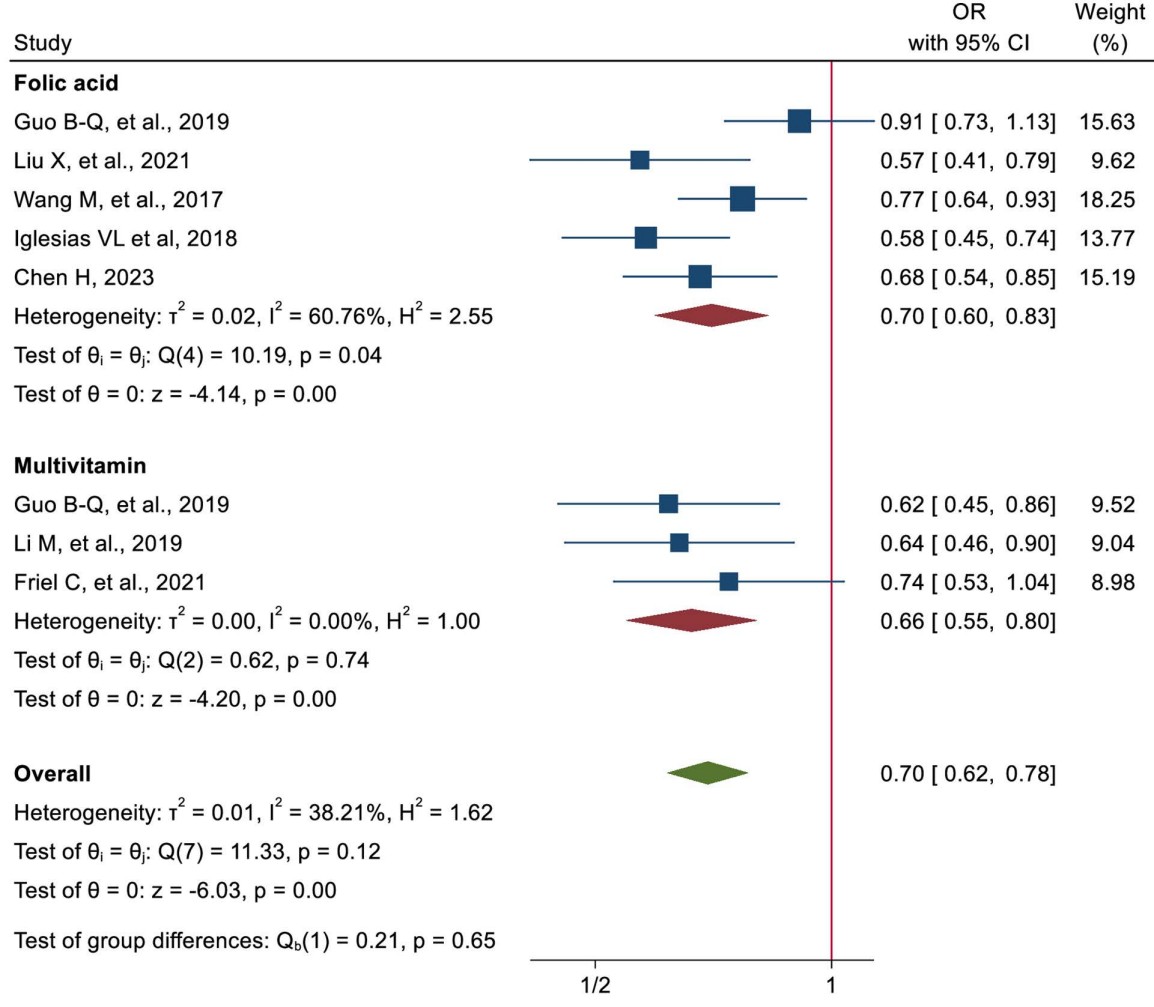

**Fig 2. Forest plot shows the pooled effect of pre-conception FA and multivitamins for the prevention of autism.**

## Publication bias assessment

The symmetrical funnel plots and Egger's test provided no evidence of substantial publication bias among the included studies (P-value = 0.14).

## Credibility according to GRADE evidence assessment

Overall, the strength of evidence assessed using the GRADE framework indicates highly suggestive (Class II) evidence for both the pooled analysis of the effects of maternal prenatal folic acid and multivitamin supplementation in preventing ASD. Both pooled analyses met most of the GRADE criteria. Specifically, both analyses demonstrated low to moderate heterogeneity, with I² values of 0.00% (P = 0.74) and 60.76% (P = 0.04), respectively. No publication bias was detected in either analysis (Egger's test, P = 0.09). Furthermore, both analyses met the other three GRADE criteria: 1) no serious indirectness, 2) no serious imprecision, and 3) low risk of bias assessment using AMSTAR (A Measurement Tool to Assess Systematic Reviews), where 75% of questions were answered 'yes' and 25% were rated as 'unclear' or 'no.' As a result,

the overall certainty of the evidence for both folic acid and multivitamin supplementation in preventing ASD highly suggests a 30% reduction in the risk of ASD for folic acid supplementation and a 34% reduction in the risk of ASD for multivitamin supplementation (S2 Table).

## Discussion

In this umbrella review, we pooled the data from eight SRMs, and the results of the overall analysis indicated a lower risk of ASD among children whose mothers took prenatal folic acid and multivitamin supplements, compared to those children whose mothers did not take the supplements.

Our finding suggests that prenatal folic acid and multivitamin supplementation may play a protective role in reducing the risk of ASD in offspring. Several potential mechanisms could explain this association. Folic acid, a vital nutrient during pregnancy, is involved in DNA methylation and neural tube development, which are crucial for proper brain development [28–30]. Adequate folate levels during early pregnancy may help prevent neural defects that could contribute to neurodevelopmental disorders including ASD. Several existing studies have reported a link between reduced folate levels and an increased risk of neural tube defects [31–33]. Additionally, folic acid may play a role in modulating inflammation and oxidative stress [34], both of which have been implicated in the development of ASD. During neurogenesis and cell migration, high concentrations of methyl donors are required in most cortical and subcortical structures [35]. Studies have examined the possible causative relationship between folate intake during preconception and pregnancy its metabolism, and the onset of ASDs, given the role folate plays in the developing brain through DNA synthesis, neurotransmitter production, and myelination [10–16,27]. However, the results remain conflicting. The timing of maternal folate intake appears to be a critical factor; a child's risk of autism was reduced only when the supplements were taken preconceptionally and during early pregnancy, specifically between 4 weeks before and eight weeks after the start of pregnancy [36].

The current review explored maternal exposure to multivitamins and folic acid supplementation during pregnancy and its association with ASD in offspring. However, identifying the specific ingredient(s) responsible for this protective effect remains challenging, as most studies did not provide detailed information on the composition or quantities of individual vitamins and minerals within the multivitamins used. Nutrients are vital environmental factors that play a crucial role in embryogenesis, fetal growth, and neurobiological development [37]. Multivitamins are among the most common dietary supplements, typically containing a combination of essential vitamins and minerals [38]. Among these, vitamin B9 (folate) is well-documented for its critical roles in neuronal differentiation, maturation, neurotrophic functions, and neuroprotection, alongside its traditional function in calcium and phosphorus metabolism. These attributes suggest folate may contribute significantly to the observed protective effects. Additionally, other studies have suggested that certain nutrients, such as zinc and vitamin D, may also play a role in the nutritional management of autism [39,40]. Vitamin D, similar to folate, is essential for neuronal differentiation, maturation, neurotrophic functions, and neuroprotection [41]. Beyond these roles, it is well known for its involvement in calcium and phosphorus regulation [42]. Zinc, although less discussed in the context of multivitamin supplementation, is recognized for its importance in brain development and function, making it a potential candidate for further exploration [40].

Folic acid deficiency before and during early pregnancy is a well-established risk factor for NTDs and may contribute to brain abnormalities associated with ASD [43–46]. Periconceptional folic acid supplementation has been shown to reduce NTD risk by up to 70% [47]. While a daily dose of 400 µg of folic acid is recommended to prevent NTDs, its role in reducing ASD risk remains complex and influenced by other factors [12]. Although emerging evidence suggests maternal folic acid supplementation may lower ASD risk, further rigorous population-based studies are needed to identify the most effective timing and dosage [47]. Understanding these factors is crucial for developing targeted strategies to prevent neurodevelopmental disorders.

## Strengths and limitations of the study

Several limitations should be acknowledged. First, the included systematic reviews and meta-analyses did not cover all global regions, which may limit the generalizability of the findings. Second, mild to moderate heterogeneity persisted

despite efforts to address it through subgroup analyses and random-effects modeling. Third, variations in supplementation timing, dosage, and formulation across studies posed challenges for interpretation. Finally, some reviews lacked protocol registration and did not consistently assess risk of bias or publication bias, potentially affecting the reliability of their conclusions.

Despite these limitations, this umbrella review advances the literature by jointly analysing both folic acid and multivitamin supplementation, providing a more comprehensive synthesis of their potential impact on ASD risk. The study adhered to PRISMA guidelines, ensuring methodological rigor, and evaluated the quality of included reviews using the AMSTAR tool [48–51]. Subgroup analyses by supplement type further enhanced the robustness of the findings.

### Implications for practice, policy, and future research

Practice: The findings emphasize the importance of prenatal folic acid and multivitamin supplementation as part of routine maternal care to reduce the risk of ASDs in offspring. Healthcare providers should encourage women to start supplementation during the pre-conception period and continue throughout pregnancy. This proactive approach can be integrated into preconception and prenatal care programs to promote maternal and fetal health.

Policy: Policymakers should prioritize the inclusion of folic acid and multivitamin supplementation in national maternal health guidelines and public health initiatives. Public awareness campaigns are needed to educate women of reproductive age about the potential benefits of these supplements in reducing ASD risk. Subsidizing or ensuring the affordability of high-quality folic acid and multivitamin supplements, particularly in low-resource settings, can further enhance accessibility and adherence to these recommendations.

Research: Future research should focus on conducting large-scale prospective cohort studies and randomized controlled trials (RCTs) to confirm and expand these findings. Key areas of investigation include exploring the molecular mechanisms behind the protective effects of folic acid and multivitamins against ASD, determining the optimal dosage, duration, and timing of maternal supplementation for maximum risk reduction, and assessing the roles of specific nutrients like vitamin D, zinc, and folate in mitigating ASD risk. Additionally, it is important to evaluate the impact of maternal supplementation across diverse populations and geographic regions to enhance generalizability and ensure culturally relevant interventions. Addressing these gaps will strengthen evidence to refine clinical guidelines and improve preventive strategies for ASD.

### Conclusion and recommendation

This umbrella review found that maternal prenatal folic acid or multivitamin supplementation was associated with a reduced risk of ASD in children. The evidence is highly suggestive, as assessed using the GRADE approach, indicating strong support for the protective effects of these supplements. Incorporating prenatal multivitamin and folic acid supplementation into ASD prevention strategies is recommended during the pre-conception and early pregnancy period. Further large-scale studies and RCTs are needed to clarify their protective effects, underlying mechanisms, and optimal dosage, timing, and duration for reducing ASD risk.

### Supporting information

**S1 Fig. Sensitivity analysis for the effect of pre-conception FA and multivitamins for the prevention of autism.**
(TIF)

**S2 Fig. Funnel plot for the effect of pre-conception FA and multivitamins for the prevention of autism.**
(TIF)

**S1 Table. Search strategy used for one of the databases.**
(DOCX)

**S2 Table. Shows quality appraisal of included SRM using AMSTAR-2 checklist.**
(DOCX)

**S3 Table. Quality assessment using the GRADE framework of each pooled analysis.**
(DOCX)

**S4 Table. PRISMA_2020_checklist.**
(DOCX)

**S5 Table. Descriptions of 667 articles.**
(XLSX)

## Author contributions

**Conceptualization:** Biruk Abate, Desie Temesgen, Kindie Mekuria, Addis Wondmagegn Alamaw, Molla Azmeraw, Alemu Birara Zemariam, Tegene Atamenta Kitaw, Amare Kassaw, Befkad Derese Tilahun, Molalign Aligaz Adisu.

**Data curation:** Biruk Abate, Biruk Shalmeno Tusa, Ashenafi Kibret Sendekie, Desie Temesgen, Addis Wondmagegn Alamaw, Molla Azmeraw, Tegene Atamenta Kitaw, Befkad Derese Tilahun, Gizachew Yilak, Molalign Aligaz Adisu, Berihun Dachew.

**Formal analysis:** Biruk Abate, Biruk Shalmeno Tusa, Berihun Dachew.

**Funding acquisition:** Biruk Abate.

**Investigation:** Biruk Abate, Alemu Birara Zemariam, Amare Kassaw, Gebremeskel Kibret Abebe, Befkad Derese Tilahun, Gizachew Yilak, Molalign Aligaz Adisu.

**Methodology:** Biruk Abate, Biruk Shalmeno Tusa, Ashenafi Kibret Sendekie, Desie Temesgen, Addis Wondmagegn Alamaw, Molla Azmeraw, Alemu Birara Zemariam, Amare Kassaw, Gebremeskel Kibret Abebe, Befkad Derese Tilahun, Gizachew Yilak, Molalign Aligaz Adisu, Berihun Dachew.

**Project administration:** Biruk Abate.

**Resources:** Biruk Abate, Kindie Mekuria.

**Software:** Biruk Abate, Biruk Shalmeno Tusa, Berihun Dachew.

**Supervision:** Biruk Abate, Biruk Shalmeno Tusa, Gizachew Yilak, Berihun Dachew.

**Validation:** Biruk Abate, Biruk Shalmeno Tusa, Ashenafi Kibret Sendekie, Kindie Mekuria, Addis Wondmagegn Alamaw, Alemu Birara Zemariam, Tegene Atamenta Kitaw, Gebremeskel Kibret Abebe, Berihun Dachew.

**Visualization:** Biruk Abate, Biruk Shalmeno Tusa, Ashenafi Kibret Sendekie, Desie Temesgen, Kindie Mekuria, Addis Wondmagegn Alamaw, Molla Azmeraw, Alemu Birara Zemariam, Tegene Atamenta Kitaw, Amare Kassaw, Gebremeskel Kibret Abebe, Befkad Derese Tilahun, Gizachew Yilak, Molalign Aligaz Adisu, Berihun Dachew.

**Writing – original draft:** Biruk Abate, Biruk Shalmeno Tusa, Ashenafi Kibret Sendekie, Desie Temesgen, Kindie Mekuria, Addis Wondmagegn Alamaw, Molla Azmeraw, Alemu Birara Zemariam, Tegene Atamenta Kitaw, Amare Kassaw, Gebremeskel Kibret Abebe, Befkad Derese Tilahun, Gizachew Yilak, Molalign Aligaz Adisu, Berihun Dachew.

**Writing – review & editing:** Biruk Abate, Biruk Shalmeno Tusa, Ashenafi Kibret Sendekie, Desie Temesgen, Kindie Mekuria, Addis Wondmagegn Alamaw, Molla Azmeraw, Alemu Birara Zemariam, Tegene Atamenta Kitaw, Amare Kassaw, Gebremeskel Kibret Abebe, Befkad Derese Tilahun, Gizachew Yilak, Molalign Aligaz Adisu, Berihun Dachew.

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
