## [Decision Letter · Decision Letter 0]

6 Jan 2025

Dear Dr.  Abate,

Thank you for submitting your manuscript to PLOS ONE. After careful consideration, we feel that it has merit but does not fully meet PLOS ONE’s publication criteria as it currently stands. Therefore, we invite you to submit a revised version of the manuscript that addresses the points raised during the review process.

**comments/suggestion to author;**
**In order to published this manuscript, you have to dress  the two reviewers comments one by one. The two reviewers addressed and critical reviewed  the manuscript so that there is no conflict between them. Finally I  need changes in this manuscript, grammatically, throughout the manuscript, and in the discussion part, you discussed only folic acid effects but  no multivitamins .**

We look forward to receiving your revised manuscript.

Kind regards,

Engidaw Fentahun Enyew

Academic Editor

PLOS ONE

Journal Requirements:

2. We note that your Data Availability Statement is currently as follows: “All relevant data are within the manuscript and in Supporting Information files.”

3. Please include captions for your Supporting Information files at the end of your manuscript, and update any in-text citations to match accordingly. Please see our Supporting Information guidelines for more information: http://journals.plos.org/plosone/s/supporting-information .

4. As required by our policy on Data Availability, please ensure your manuscript or supplementary information includes the following:

Additional Editor Comments:

No more additional comments 

Reviewers' comments:

Reviewer's Responses to Questions

**Comments to the Author**

1. Is the manuscript technically sound, and do the data support the conclusions?

Reviewer #1: Partly

Reviewer #2: Yes

2. Has the statistical analysis been performed appropriately and rigorously?

Reviewer #1: Yes

Reviewer #2: Yes

3. Have the authors made all data underlying the findings in their manuscript fully available?

Reviewer #1: Yes

Reviewer #2: Yes

4. Is the manuscript presented in an intelligible fashion and written in standard English?

Reviewer #1: No

Reviewer #2: Yes

Reviewer #1: Pre-conception folic acid and multivitamin supplementation for prevention of autism in offspring: An umbrella review of Systematic Review and Meta-analysis

Thank you for the opportunity to review this umbrella review. It addresses an important question, however it is not effective in the message it endeavors to deliver. There are some major grammatical and formatting errors throughout, specifically the results tables. I am not convinced that this manuscript adds to what is already known regarding supplementation and outcomes of ASD. I believe there could be more substantial analysis, and a greater discussion regarding the results and the implications.

Abstract

Result : please explain SRM.

‘Subgroup analysis by type of supplement revealed that the pooled effect of either folic acid or multivitamin supplementation for the prevention of autism was found to be 0.69(0.58, 0.80) in the multivitamin group, while this estimate is 0.70(0.55,0.85) among studies with folic acid supplementation.’ Please give units/meaning to this figures, and rephrase this to avoid such repetition, for example Subgroup analysis by type of supplement revealed that the pooled effect of either folic acid alone or multivitamin supplementation for the prevention of autism was found to be 0.70(0.55,0.85) and 0.69(0.58, 0.80) respectively.

‘with a slightly higher (1%) percentage of reduction in autism’ please rephrase to: with a 1% greater reduction in autism’.

Introduction

‘Ongoing studies are most frequently considered in environmental risk factors’ can you reference this please?

‘Preconception and prenatal maternal nutrition are among the modifiable risk factors of ASD’ please add a reference.

‘The current review explored that during pregnancy, maternal exposure to Multivitamins and folic acid supplementation significantly reduces the likelihood of ASD in offspring.’ Please rephrase – the current review explores maternal exposure to multivitamins and folic acid supplementation during pregnancy and its association with ASD in offspring’ or something similar.

‘Folic is water soluble’ please add folic acid.

‘Various folate forms participate in essential reactions, like DNA methylation and replication’ – needs a reference.

‘FA is metabolized differently to naturally occurring forms of folates with a bioavailability of approximately 70% higher. 7’ Please also rephrase this sentence as it is not clear what the meaning of it is.

Folic acid is not abbreviated to FA prior to this – please use folic acid (FA) at first use, and then continue to use FA throughout. Please be consistent in either using the full term of folic acid or the abbreviation FA.

‘Since folate is essential for metabolism and nervous system function and as a precursor of S-adenosyl-methionine (SAM), the universal methyl donor that is essential for cellular methylation of lipids, proteins, nucleic acid, and metabolites, inadequate availability affects the gene expression in the neurodevelopmental process which may have an association between maternal folate level during pregnancy and ASD mediated by DNA methylation. 8, 9’ please shorten this sentence or break it into multiple sentences. It is quite long as it is and difficult to follow for the reader.

Methods

SRM is introduced without full explanation – please write the term in full before using it as an abbreviation.

‘From checking PROSPERO, this umbrella review needed to be registered.’ Please remove.

‘We searched systematic review and meta-analysis’ please rephrase to We searched for systematic reviews and meta-analyses

‘in global context on 23/10/ 2023 G. C. Were searched SRM which reported the effect of preconception folic acid and/or multivitamin supplementation for prevention of autism in global context using PICO frameworks’ please rephrase as this is incorrect grammar and difficult to decipher what you are saying, specifically at the mention of G.C. and we ‘searched SRM which reported…’

PICO only needs to be mentioned once.

‘was also performed’ remove also.

‘The quality scoring was done out of 11, with scores 8–11, 4–7, and < 3 indicating high, medium, and low qualities’ Please re-organize this sentence to start from low, medium and high so that it reads chronologically.

There is only mention of investigators screening the title and abstracts, what about the full text screening? How were disagreements resolved?

‘After the data was extracted using Microsoft Excel format, we imported the data to STATA version 17.0 statistical software for further analysis’ please reduce this to ‘STATA v17.0 was used for statistical analyses.

Results

‘After duplication was removed’ please correct the grammar so that it reads ‘ after duplicates were removed’.

The flow chart needs clarification – Were the 721 duplicates? Or that’s how many remained after duplicate removal? It needs to be cleaned in terms of layout also – the boxes are not in line, the arrows are not in line, the text in the blue box doesn’t fit. It needs to be cleaned.

Please highlight further the ‘quality reasons’ for exclusion in your methods.

‘Included SRM included from five to 16 studies’ please rephrase to avoid repetition of included, it is confusing.

‘….0.57, 95% CI: 0.41, 0.78) 15 to 0.91, 95% CI: 0.73, 1.13)’ please fix the use of brackets.

‘The random-effects model analysis from those studies revealed that, the pooled effect of either folic acid or multivitamin supplementation’ remove the comma or rephrase please.

‘This indicates prenatal folic acid and /or folic acid supplementation was associated’ please correct this – there is no distinction between the and/or folic acid mentioned.

‘Subgroup analysis was done through stratified by the type of supplementation. Based on this..’ please rephrase. Remove ‘Based on this’.

‘Based on this, the pooled effect of either folic acid or multivitamin supplementation for the prevention of autism was found to be 0.69(0.58, 0.80) in in multivitamin group, while this estimate is 0.70(0.55,0.85) among studies with folic acid supplementation’ remove the second ‘in’ and rephrase so that each supplementation type is only mentioned once – there is too much repetition. Please give meaning to the figures – there is no unit or indication as to what these figures represent.

‘which indicated that, the absence of publication bias’ rephrase to ‘which indicates the absence of publication bias’.

It is evident from your results that you are using screenshots from the STATA output of results. Please enter these into appropriately formatted tables, only highlighting the necessary information as opposed to the excess of information and the many decimal points from the screenshot. It is inappropriate.

‘Our pooled estimated effect of preconception folic acid and multivitamin for the prevention of autism varied from 0.68 (0.57– 0.78) to 0.71 (0.62–0.81) after the deletion of a single study (Fig 5).’ Use the full-term figure as opposed to fig, and give meaning to the numbers.

I think it would be interesting to ascertain if the multivitamins used in the studies contained folic acid. The introduction highlighted the lack of consensus regarding dose, frequency and timing of supplementation, but this was not addressed in the results.

There is little mentioned regarding the quality of the reviews included in the results. Please rectify.

As this is a global context, I think it would be interesting to see exactly what countries were included. I would imagine a difference would occur depending on socioeconomic status of the countries included, among other environmental factors.

Discussion

‘increases the risk of (NTDs)’ please include Neural Tube Defects.

‘insufficient folic acid intake during the early stages of pregnancy can result in (NTDs),’ remove brackets around NTDs.

Your discussion is brief, and focuses only on folic acid, when the review includes multivitamins. Please consider the numerous aspects at play here, the discussion to me does not consider all of the evidence and is very one-sided. Please broaden.

Is 7 studies considered substantial? Particularly when you started with >1000. I would have noted the low number of studies included.

You mention AMSTAR is used but there is no evidence of this use in your results.

‘Despite these strengths the study also has few limitations: as the included studies were not from all countries and this may affect the generalizability of the pooled result’ this should be rephrased to improve grammar.

‘despite the authors tried to reduce it through’ please rephrase to ‘despite the authors efforts to reduce it…’

‘Participants who took multivitamin supplementation were associated with a slightly higher (1%) percentage of reduction in autism compared with those who took folic acid.’ Rephrase as highlighted previously.

I am not convinced that this umbrella review really adds to what is already known. Please elaborate in your discussion, highlighting the importance of this work and how it addresses the gaps in our existing knowledge. I feel as though the introduction leads the reader to believe that this will look at importance dose quantity, frequency, supplementation timing, etc., which is not the case.

Reviewer #2: Abstract

IMPORTANCE: Previous reviews explored the association between maternal use of folic acid and multivitamin supplements and the risk of autism spectrum disorders (ASD) in

OBJECTIVE: To combine the inconsistent data on the effect of prenatal folic acid and multivitamin supplementation for the prevention of autism in offspring.

DESIGN, SETTING, AND PARTICIPANTS: In this umbrella review using PRISMA guideline, PubMed, Embase, Scopus, Web of Sciences, Cochrane Database of Systematic Reviews, Scopus, and Google Scholar which reported the effects of folic acid and multivitamin supplementation for the prevention of autism in offspring were searched. The quality of the included studies was assessed using the Assessment of Multiple Systematic Reviews (AMSTAR). A weighted inverse variance random-effects model was applied to find the pooled estimates. The subgroup analysis, heterogeneity, publication bias, and sensitivity analysis were also assessed.

EXPOSURES: Maternal multivitamin/folic supplements were classified for folic acid, multivitamin supplements, and combination thereof exposed in the intervals before pregnancy. MAIN OUTCOMES AND MEASURES: The association between maternal multivitamin supplementation and the risk of autism in offspring was quantified with relative risks (RRs) and their 95% Cis.

RESULT: Seven SRMs with 3,785,573 study participants were included. The pooled effect of either folic acid or multivitamin supplementation for the prevention of autism globally is found to be 0.69 (95% CI: 0.60, 0.79) (I2 = 46.4%; p <=0•083). Subgroup analysis by type of supplement revealed that the pooled effect of either folic acid or multivitamin supplementation for the prevention of autism was found to be 0.69(0.58, 0.80) in the multivitamin group, while this estimate is 0.70(0.55,0.85) among studies with folic acid supplementation.

CONCLUSION AND RELEVANCE: This umbrella review of systematic review and meta analysis found that prenatal folic acid and multivitamin supplementation was associated with a 31% reduction in autism. Participants who took multivitamin supplementation were associated with a slightly higher (1%) percentage of reduction in autism compared with those who took folic acid. This comprehensive umbrella review revealed both folic acid and multivitamins were associated with significantly lower levels of autism in children. Considering the incorporation of those supplements in autism prevention strategies during the pre-conception period is recommended. More large-scale prospective cohorts and RCTs are needed to understand the protective effect of multivitamins/ and or folic acid on ASD risk, address the molecular mechanisms, and determine the optimal dose, duration, and timing of maternal multivitamin/ and folic acid intake for best child ASD risk reduction.

I do have some comments in the abstract section

you should state introduction other than importance,

DESIGN, SETTING, AND PARTICIPANTS these are already methods

Once you stated the number in result section so you should have to state general context of your finding in conclusion for example you got 69%( result) your conclusion might be (high or low based specific guideline to say high or low)

Once in your result section you got prenatal folic acid and multivitamin supplementation for the prevention of autism globally is found to be 0.69, and in the conclusion prenatal folic acid and multivitamin supplementation was associated with a 31% reduction in autism. Am confusing with these what does it mean the finding 69% and 31% in conclusion, you see both sentences revealed reduction of autism.

DESIGN, SETTING, AND PARTICIPANTS: In this umbrella review using PRISMA guideline, PubMed, Embase, Scopus, Web of Sciences, Cochrane Database of Systematic Reviews, Scopus, and Google Scholar which reported the effects of folic acid and multivitamin supplementation for the prevention of autism in offspring were searched.

Does it prisma guideline a search engine?

Key Points QUESTION: Does maternal folic acid and multivitamin supplement use before pregnancy decrease the risk of autism in offspring?

FINDINGS In this umbrella review, seven systematic reviews and meta-analyses were included with 3,785,573 participants. A statistically significant association between maternal multivitamin and folic acid supplement use before pregnancy and reduced risk of autism in their offspring was observed. MEANING A reduced risk of autism spectrum disorder in children born to women who used multivitamins and folic acid supplements before pregnancy has important public health implications; possible mechanisms include epigenetic modifications.

Is that seven systematic review with 3,785,573 participant your key findings? It is better to report the main findings that you got from 7 systematic review articles

However, evidence derived from the previous reviews on the uses of folic acid and multivitamins for the prevention of ASD is inconclusive. Some studies concluded that there is no association between maternal folic acid supplementation during the prenatal period and the risk of ASD. 10, 11 On the other hand, other reviews revealed that there is a strong association between preconception and prenatal folic acid and multivitamin supplementations and the risk of autistic disorder. 12-15 Even for those who agree on the supplementations, there are inconsistencies in the timing (preconception or during pregnancy), the types of Nutrients (folic and Multivitamins), and each nutrient's dose. Therefore, this Umbrella review aims to explore the actual association between maternal supplementations with folic acid and multivitamins pre-conceptional and during pregnancy and the risk of ASD in offspring.

Your interest is to do umbrella review, it is not elaborated more what is the important to do umbrella review, you only said “previous reviews on the uses of folic acid and multivitamins for the prevention of ASD is inconclusive.” What are the key features of umbrella review? What gaps were not answered by systematic review? You should have to clearly show the need to use umbrella review in the context of your title

All published, unpublished, and systematic reviews and meta-analyses assess the effect of pre-conceptional folic acid and multivitamin supplementation for preventing autism in the offspring. We would have included studies reported as abstracts only and those with full-text manuscripts

We searched systematic review and meta-analysis from the following databases: PubMed, Embase, Scopus, Web of Sciences, Cochrane Database of Systematic Reviews, Scopus and Google Scholar in global context on 23/10/ 2023 G. C.

Potentially eligible studies identified through database search (n=1380), Potentially eligible studies identified through other sources (n =8)

How many articles have you identified through different search engine ( i.e how much was in Pub med, scops, unpublished article…etc )

Results and Discussion:

I did not see the implications of the study for policymakers and practitioners. Please provide unique, actionable strategies to address autism based on the findings.

Ensure that the discussion goes beyond restating the results and connects them to broader public health and policy contexts.

**Do you want your identity to be public for this peer review?** For information about this choice, including consent withdrawal, please see our Privacy Policy

Reviewer #1: No

Reviewer #2: **Yes: ** Belayneh Jejaw Abate

---

## [Author Response · Author response to Decision Letter 1]

1 May 2025

Dear Khem Descatamiento,,

Thank you for your thoughtful feedback.

Please find attached a document containing signatures from all authors, confirming our agreement with the proposed authorship changes as requested.

File name:Authorship Change PONE_all signed

Kind regards,

Biruk Beletew Abate

---

## [Decision Letter · Decision Letter 1]

26 Sep 2025

Dear Dr. Abate,

Thank you for submitting your manuscript to PLOS ONE. After careful consideration, we feel that it has merit but does not fully meet PLOS ONE’s publication criteria as it currently stands. Therefore, we invite you to submit a revised version of the manuscript that addresses the points raised during the review process.

We look forward to receiving your revised manuscript.

Kind regards,

Cecilia Nwadiuto Obasi, PhD

Academic Editor

PLOS ONE

Journal Requirements:

Reviewers' comments:

Reviewer's Responses to Questions

**Comments to the Author**

Reviewer #2: (No Response)

Reviewer #3: All comments have been addressed

Reviewer #4: (No Response)

2. Is the manuscript technically sound, and do the data support the conclusions?

Reviewer #2: Yes

Reviewer #3: Yes

Reviewer #4: Yes

3. Has the statistical analysis been performed appropriately and rigorously?

Reviewer #2: Yes

Reviewer #3: Yes

Reviewer #4: No

4. Have the authors made all data underlying the findings in their manuscript fully available?

Reviewer #2: Yes

Reviewer #3: Yes

Reviewer #4: Yes

5. Is the manuscript presented in an intelligible fashion and written in standard English?

Reviewer #2: Yes

Reviewer #3: Yes

Reviewer #4: Yes

Reviewer #2: Thank you to the authors for their efforts in addressing the previous comments. That said, I have a major concern regarding the introduction section, summarized as follows:

1. Although you mention that previous findings are "inconclusive," it is important to specify why this is the case. What are the particular limitations or gaps in the existing literature that your umbrella review aims to address? Additionally, please clarify the added value of an umbrella review compared to a standard systematic review.

2. You note that some reviews report an association while others do not, but you do not discuss potential reasons for these conflicting results. Exploring these discrepancies is essential, as it forms the main rationale for conducting an umbrella review.

3. The introduction should also briefly explain what an umbrella review entails and why this approach is the most suitable method to resolve the current inconsistencies, rather than conducting another standard systematic review.

Reviewer #3: Based on the revised manuscript, the authors have made substantial improvements in both methodological clarity and narrative structure. The previously raised concerns related to the background length, clarity of methodological reporting, justification of model selection, and robustness of findings have now been comprehensively addressed.

Reviewer #4: In the Methods

Although the manuscript states that heterogeneity and publication bias were assessed, the methodology used to evaluate these aspects is not adequately described. Specifically, the authors should clarify which statistical tests or graphical methods were employed to assess heterogeneity, such as Cochrane’s Q test, I² statistic, or tau-squared, and explain how the degree of heterogeneity was interpreted. Similarly, for publication bias, it is important to specify whether funnel plots, Egger’s regression test, or other techniques were used, and how the results were interpreted.

DISCUSSION:

Better to adequately indicate the study's limitations and than statements related to strengths

This umbrella review stands apart from prior studies by jointly analyzing both folic acid and multivitamin supplementation, offering a broader synthesis of their potential impact on autism spectrum disorder (ASD) risk. However, several limitations should be noted. The included systematic reviews and meta-analyses do not represent all global regions, which may restrict the generalizability of the findings. Mild to moderate heterogeneity persists despite subgroup analyses and the use of random-effects modeling. Additionally, inconsistencies in supplementation timing, dosage, and formulation across studies introduce interpretive challenges. Some reviews lacked protocol registration and did not consistently assess risk of bias or publication bias, which may affect the reliability of their conclusions.

**Do you want your identity to be public for this peer review?** For information about this choice, including consent withdrawal, please see our Privacy Policy

Reviewer #2: **Yes: ** Belayneh Jejaw Abate

Reviewer #3: **Yes: ** Pradeep Kumar

Reviewer #4: No

---

## [Author Response · Author response to Decision Letter 2]

1 Oct 2025

PONE-D-24-14480R1

The association between maternal prenatal folic acid and multivitamin supplementation and autism spectrum disorders in offspring: An umbrella review

PLOS ONE

Cecilia Nwadiuto Obasi, PhD

Academic Editor

PLOS ONE

Dear Dr. Cecilia Nwadiuto Obasi,

Thank you for the opportunity to submit a revised version of our manuscript titled " The association between maternal prenatal folic acid and multivitamin supplementation and autism spectrum disorders in offspring: An umbrella review”. We greatly appreciate the time and effort the reviewers have invested in providing feedback and their insightful comments and valuable suggestions. We have carefully addressed all concerns in a point-by-point manner and made the necessary revisions to the manuscript. These changes have been highlighted in yellow in the revised version for your reference. Below, you will find a detailed, point-by-point response to the reviewers’ comments and concerns. All page numbers refer to the clean, revised version of the manuscript.

Kind regards,

Biruk Abate

School of Population Health, Curtin University, Australia

Email: b.abate@postgrad.curtin.edu.au

Reviewer #2

Reviewer Comment: Thank you to the authors for their efforts in addressing the previous comments. That said, I have a major concern regarding the introduction section, summarized as follows: 1. Although you mention that previous findings are "inconclusive," it is important to specify why this is the case. What are the particular limitations or gaps in the existing literature that your umbrella review aims to address? Additionally, please clarify the added value of an umbrella review compared to a standard systematic review. 2. You note that some reviews report an association while others do not, but you do not discuss potential reasons for these conflicting results. Exploring these discrepancies is essential, as it forms the main rationale for conducting an umbrella review. 3. The introduction should also briefly explain what an umbrella review entails and why this approach is the most suitable method to resolve the current inconsistencies, rather than conducting another standard systematic review.

Authors’ Response: We sincerely thank the reviewer for the insightful comments. We have now revised the introduction accordingly and clarified the distinct contribution of our work. Specifically, an umbrella review systematically synthesises evidence from multiple SRMs, assesses their methodological quality, and evaluates the overall strength and consistency of the evidence. Unlike a single SRM, this higher-level synthesis integrates and appraises all available reviews, highlights both consistencies and discrepancies, and provides an overall grading of the evidence. This approach allows us to clarify whether maternal folic acid and/or multivitamin supplementation is consistently associated with a reduced risk of ASD in offspring.

We have revised the gap statement in the introduction section as follows: “Early prevention is preferable to treatment, as no medication can address the core symptoms of ASD; available treatments focus on managing specific co-occurring conditions or symptoms. Maternal supplementation with folic acid and multivitamins before and during pregnancy represents one of the most accessible and cost-effective preventive strategies. To date, seven systematic reviews and meta-analyses (SRMs) comprising 101 primary studies have examined whether maternal prenatal folic acid supplementation reduces the risk of ASD in offspring. However, findings from previous reviews remain inconclusive. While some reviews have reported the association between prenatal folic acid and multivitamin supplementation and a lower risk of ASD, others have found no association. However, findings remain inconclusive due to several limitations in the existing literature, including variability in study design and population characteristics, differences in the definition and measurement of supplementation exposures (folic acid alone versus multivitamins, timing, dose, and duration), heterogeneity in ASD diagnostic methods (ranging from clinical diagnoses to parent-reported outcomes), and potential publication or reporting bias. These limitations have contributed to conflicting findings: while some SRMs report a protective effect of maternal folic acid and/or multivitamin supplementation (10-14), others find no significant association (15, 16). These inconsistencies underscore the need for a more comprehensive synthesis. An umbrella review, unlike a single SRM, offers a higher-level overview by integrating findings across multiple SRMs, critically appraising their methodological quality, and grading the overall strength of the evidence. Accordingly, this umbrella review aimed to synthesise these inconsistent findings to clarify the potential role of prenatal folic acid and multivitamin supplementation in reducing the risk of ASD in offspring. (Page 3& 4: Line 72-93)”

Reviewer #3:

Reviewer Comment: Based on the revised manuscript, the authors have made substantial improvements in both methodological clarity and narrative structure. The previously raised concerns related to the background length, clarity of methodological reporting, justification of model selection, and robustness of findings have now been comprehensively addressed.

Authors’ Response: We sincerely thank Reviewer #3 for the constructive feedback, which has significantly improved the clarity and quality of the manuscript.

Reviewer #4

Reviewer Comment: Although the manuscript states that heterogeneity and publication bias were assessed, the methodology used to evaluate these aspects is not adequately described. Specifically, the authors should clarify which statistical tests or graphical methods were employed to assess heterogeneity, such as Cochrane’s Q test, I² statistic, or tau-squared, and explain how the degree of heterogeneity was interpreted. Similarly, for publication bias, it is important to specify whether funnel plots, Egger’s regression test, or other techniques were used, and how the results were interpreted.

Authors’ Response: We sincerely thank the reviewer for highlighting the need for greater clarity regarding our assessment of heterogeneity and publication bias. In response, we have now revised the methodology section to explicitly describe the statistical tests and graphical methods employed. Specifically, we used Cochrane’s Q test, the I² statistic, and tau-squared (τ²) to assess heterogeneity, and we interpreted the degree of heterogeneity following established thresholds (I² values of 25%, 50%, and 75% representing low, moderate, and high heterogeneity, respectively). For publication bias, we employed funnel plots and Egger’s regression test, with asymmetry in the funnel plot and a p-value <0.05 in Egger’s test considered suggestive of potential bias. We have also clarified how these results were interpreted in the context of our findings (See Methods, Page 7&8, Lines 164-175).

We have revised the gap statement in the Methods section as follows:

We assessed heterogeneity across studies using Cochrane’s Q test (Chi-square), the I² statistic, tau-squared (τ²), and corresponding p-values (22). Heterogeneity was interpreted according to conventional thresholds, with I² values of 0% indicating no heterogeneity and values of 25%, 50%, and 75% representing low, moderate, and high heterogeneity, respectively (20, 23). For analyses where significant heterogeneity was detected, we applied the DerSimonian–Laird random-effects model. Subgroup analyses were conducted according to supplement type (folic acid versus multivitamin). To evaluate the robustness of findings, sensitivity analyses were performed by sequentially excluding individual studies. Publication bias was assessed using both graphical and statistical approaches: visual inspection of funnel plot symmetry and Egger’s regression test. Funnel plot asymmetry and a p-value <0.05 in Egger’s test were considered indicative of potential publication bias (24).

DISCUSSION:

Reviewer Comment: Better to adequately indicate the study's limitations and then statements related to strengths. This umbrella review stands apart from prior studies by jointly analyzing both folic acid and multivitamin supplementation, offering a broader synthesis of their potential impact on autism spectrum disorder (ASD) risk. However, several limitations should be noted. The included systematic reviews and meta-analyses do not represent all global regions, which may restrict the generalizability of the findings. Mild to moderate heterogeneity persists despite subgroup analyses and the use of random-effects modeling. Additionally, inconsistencies in supplementation timing, dosage, and formulation across studies introduce interpretive challenges. Some reviews lacked protocol registration and did not consistently assess risk of bias or publication bias, which may affect the reliability of their conclusions.

Authors’ Response: We thank the reviewer for the insightful comments. We have now revised the discussion section accordingly (see discussion, Pages 12&13, Lines 296-308)

We have revised the limitation statements in the Discussion section as follows:

Strengths and Limitations of the Study

Several limitations should be acknowledged. First, the included systematic reviews and meta-analyses did not cover all global regions, which may limit the generalizability of the findings. Second, mild to moderate heterogeneity persisted despite efforts to address it through subgroup analyses and random-effects modeling. Third, variations in supplementation timing, dosage, and formulation across studies posed challenges for interpretation. Finally, some reviews lacked protocol registration and did not consistently assess risk of bias or publication bias, potentially affecting the reliability of their conclusions.

Despite these limitations, this umbrella review advances the literature by jointly analysing both folic acid and multivitamin supplementation, providing a more comprehensive synthesis of their potential impact on ASD risk. The study adhered to PRISMA guidelines, ensuring methodological rigor, and evaluated the quality of included reviews using the AMSTAR tool. Subgroup analyses by supplement type further enhanced the robustness of the findings.

---

## [Editor Report · Decision Letter 2]

3 Oct 2025

The association between maternal prenatal folic acid and multivitamin supplementation and autism spectrum disorders in offspring: An umbrella review

PONE-D-24-14480R2

Dear Dr. Abate,

We’re pleased to inform you that your manuscript has been judged scientifically suitable for publication and will be formally accepted for publication once it meets all outstanding technical requirements.

Kind regards,

Cecilia Nwadiuto Obasi, PhD

Academic Editor

PLOS ONE
---

## [Editor Report · Acceptance letter]

PONE-D-24-14480R2

PLOS ONE

Dear Dr. Abate,

I'm pleased to inform you that your manuscript has been deemed suitable for publication in PLOS ONE. Congratulations! Your manuscript is now being handed over to our production team.

Kind regards,

on behalf of

Dr. Cecilia Nwadiuto Obasi

Academic Editor

PLOS ONE